# IL-1β Implications in Type 1 Diabetes Mellitus Progression: Systematic Review and Meta-Analysis

**DOI:** 10.3390/jcm11051303

**Published:** 2022-02-27

**Authors:** Fátima Cano-Cano, Laura Gómez-Jaramillo, Pablo Ramos-García, Ana I. Arroba, Manuel Aguilar-Diosdado

**Affiliations:** 1Research Unit, Biomedical Research and Innovation Institute of Cadiz (INiBICA), Puerta del Mar University Hospital, 11009 Cadiz, Spain; canocano.fatima@gmail.com (F.C.-C.); laugomjar@gmail.com (L.G.-J.); manuel.aguilar.sspa@juntadeandalucia.es (M.A.-D.); 2Faculty of Dentistry, University of Granada, 18011 Granada, Spain; 3Department of Endocrinology and Metabolism, University Hospital Puerta del Mar, 11009 Cadiz, Spain

**Keywords:** IL-1β, type 1 diabetes mellitus, chronic inflammation, systematic review, meta-analysis

## Abstract

During Type 1 Diabetes Mellitus (T1DM) progression, there is chronic and low-grade inflammation that could be related to the evolution of the disease. We carried out a systematic review and meta-analysis to evaluate whether peripheral levels of pro-inflammatory markers such as interleukin-1 beta (IL-1β) is significantly different among patients with or without T1DM, in gender, management of the T1DM, detection in several biological fluids, study design, age range, and glycated hemoglobin. We searched PubMed, Embase, Web of Science, and Scopus databases, and 26 relevant studies (2186 with T1DM, 2047 controls) were included. We evaluated the studies’ quality using the Newcastle–Ottawa scale. Meta-analyses were conducted, and heterogeneity and publication bias were examined. Compared with controls, IL-1β determined by immunoassays (pooled standardized mean difference (SMD): 2.45, 95% CI = 1.73 to 3.17; *p* < 0.001) was significantly elevated in T1DM. The compared IL-1β levels in patients <18 years (SMD = 2.81, 95% CI = 1.88–3.74) was significantly elevated. The hemoglobin-glycated (Hbg) levels in patients <18 years were compared (Hbg > 7: SMD = 5.43, 95% CI = 3.31–7.56; *p* = 0.001). Compared with the study design, IL-1β evaluated by ELISA (pooled SMD = 3.29, 95% CI = 2.27 to 4.30, *p* < 0.001) was significantly elevated in T1DM patients. IL-1β remained significantly higher in patients with a worse management of T1DM and in the early stage of T1DM. IL-1β levels determine the inflammatory environment during T1DM.

## 1. Introduction

Type 1 diabetes mellitus (T1DM) is an autoimmune disease often diagnosed in childhood that progresses with pancreatic β-cell destruction and life-long insulin dependence. T1DM susceptibility involves a complex interplay between genetic and environmental factors and with the participation of adaptive immunity, although there is now growing evidence for the role of innate inflammation [1].

T1DM, in the early stage of the disease, is characterized by chronic inflammation that involves pancreatic islet degeneration. The maintenance activation of the innate immune system impairs insulin secretion and action, and inflammation also contributes to diabetes complications, such as diabetic retinopathy and nephropathy. Prior to the manifestation of the disease, a pre-diabetic period may last several years and is characterized by the detection of circulating autoantibodies against beta-cell antigens [2]. There is evidence that indicates a direct pathogenic effect of IL-1β on the islet during the development of T1DM. In pancreatic samples from adult living donors, the presence of IL-β and TNF-α has been detected, mainly in macrophages and dendritic cells [3]. However, despite strong preclinical evidence demonstrating that targeting inflammatory pathways can prevent secondary complications, there are still no treatments for diabetes that target innate immune mediators [4].

In patients with T1DM, higher levels of proinflammatory cytokines have been detected that are physiological constituents of any inflammatory reaction, including interleukins (IL-1α, IL-1β, IL-10, IL-12), interferons (IFNα/β, IFNγ), transforming growth factor-β (TGF-β), tumor necrosis factors (TNFα, TNFβ), and nitric oxide (NO) [5]. The microenvironment is also enriched in anti-inflammatory cytokines including IL-4, IL-10, IL-13, and IL-22, which are generally associated with protective effects over β-cell survival [6].

The role of cytokines in the pathogenesis of autoimmune disorders, particularly T1DM, has been extensively investigated to determine their potential therapeutic value. Screening for the presence of cytokines during the early stages of T1DM can serve to identify immunological response-related soluble factors and a better diagnosis and treatment of the disease.

Interleukin 1 (IL-1) is a 17 kDa protein highly conserved through evolution and is a key mediator of inflammation [7], and it has been suggested as candidate for inducing beta-cell apoptosis in vitro and aggravating diabetes in vivo. Recently, a significant number of studies have given attention to the role of IL-1β in the pathogenesis of autoimmune and inflammatory diseases. There are numerous studies that relate the polymorphisms and gene variations in the IL-1β gene with the differences in the transcription and expression of the IL-1β gene that could correlate with the development of many autoimmune and inflammatory diseases, such as systemic lupus erythematosus [8], rheumatoid arthritis [9], and multiple sclerosis [10].

The genetic or pharmacological inhibition of IL-1 action has clinical efficacy in many inflammatory diseases, due to IL-1 acting on T-lymphocyte regulation. The adverse effects of IL-1β on human beta cells in vitro and in animal models have promoted recent clinical trials in volunteers with recent-onset type 1 diabetes, using strategies involving the systemic blockade of IL-1β or its receptors [7,11]. Genetic or pharmacological abrogation of IL-1 action reduces disease incidence in animal models of type 1 diabetes mellitus [12]. The modulating effect of IL-1 on the interaction between the innate and adaptive immune systems and the effects of IL-1 on the beta-cell point to this molecule being a potential interventional target in autoimmune diabetes mellitus.

Regarding the participation of other pro-inflammatory cytokines, such as TNF-α and IL6, in patients with T1DM, these were found to be linked to elevated level of serum IL-6 and TNF-α, on which the age, ethnicity, and disease duration [12,13] in T1DM patients had no effect on the serum IL-6 levels for promoting diabetes mellitus. The IL-1β level’s modulation during different stages of T1DM could be a sensor of progression and good management of disease over time.

With this background, we conducted the first systematic review and meta-analysis to qualitatively and quantitatively evaluate the available scientific evidence on circulating IL-1β levels in T1DM. The aim of this work is to determine the modulated levels of IL-1β between patients with or without T1DM, and to explore their hypothetical influential variables (i.e., geographical area, age, sex, human tissues, biochemical parameters, research methods, and IL-1β determination techniques).

## 2. Material and Methods

This systematic review and meta-analysis complied with *Preferred Reporting Items for Systematic Reviews and Meta-Analyses* (PRISMA) and *Meta-analysis of Observational Studies in Epidemiology* (MOOSE) guidelines [14,15], and closely followed the criteria of the *Cochrane Handbook for Systematic Reviews of Interventions* [16].

### 2.1. Protocol

In order to minimize the risk of bias and improve the transparency, precision, and integrity of this study, a protocol on its methodology was a priori registered in the PROSPERO international prospective register of systematic reviews (www.crd.york.ac.uk/PROSPERO, accessed on July 2020, registration code CRD42020180062) [17]. The protocol adhered to the PRISMA-P statement to ensure a rigorous approach [18].

### 2.2. Search Strategy

We searched PubMed, Embase, Web of Science, and Scopus databases for studies published before the search date (upper limit = October 2020), with no lower date limit. The searches were conducted by combining thesaurus terms used by the databases (i.e., MeSH and EMTREE) with free terms (Appendix A), and built to maximize sensitivity. We also manually screened the reference lists of retrieved studies for additional relevant studies. All references were managed using Mendeley Desktop v. 1.19.4 (Elsevier, Amsterdam, The Netherlands); duplicate references were eliminated using this software.

### 2.3. Eligibility Criteria

Inclusion criteria: (1) original research studies without language, publication date, follow up periods, study design, geographical area, sex, or age restrictions; (2) T1DM subjects compared to no T1DM as control group; (3) lL-1β determination by enzyme-linked immunosorbent assay (ELISA), quantitative real time polymerase chain reaction (qRT-PCR) and/or flow cytometry in human samples from any anatomical origin; (4) the names and affiliations of authors, recruitment period and settings were examined to determine whether studies were conducted in the same study population. In such cases, we included the most recent study or that which published more complete data.

Exclusion criteria: (1) retracted articles, reviews, meta-analyses, case reports, clinical trials, editorials, letters, abstracts of scientific meetings, personal opinions or comments, and book chapters; (2) in vitro and animal experimental studies; (3) studies that do not report the disease of interest (i.e., T1DM), do not assess IL-1β levels, or those without a control group; (4) studies reporting insufficient data to extract or estimate mean ± standard deviation (SD); (5) overlapping populations.

### 2.4. Study Selection Process

The eligibility criteria were applied independently by three authors (LGJ, FCC, and AIA). Discrepancies were resolved by consensus with a fourth author (PRG). Articles were selected in two phases, first screening the titles and abstracts of the retrieved articles in an initial selection, and then reading the full text of the selected articles, excluding those that did not meet the review eligibility criteria.

### 2.5. Data Extraction

Three authors (LGJ, FCC, and AIA) independently extracted data from the selected articles, completing a data collection form in a standardized manner using Excel v. Microsoft Office Professional Plus 2013 (Microsoft. Redmond, WA, USA). These data were additionally cross-checked in multiple rounds, solving discrepancies by consensus. The data were gathered on the first author, publication year, study country and continent, sample size, source of sample (i.e., type of tissue), IL-1β determination (extracting means ± SD and measuring units) in T1DM and controls (i.e., patients not affected by T1DM), age, year of diagnosis, sex of patients, Hbg levels, research methods analysis technique (e.g., ELISA or qRT-PCR), and type of study (i.e., cross-sectional, case-control, or cohorts).

### 2.6. Evaluation of Quality and Risk of Bias

We used the Newcastle–Ottawa quality assessment scale (NOS) to assess the risk of bias [19]. The evaluation was conducted by two independent reviewers who were knowledgeable about the content and methodology. The results were compared and conflicts resolved by agreement between the two reviewers, with input from a third reviewer if necessary. The studies that received a star in each domain were considered to be of high quality. The maximum score was 8, the minimum 0. It was decided a priori that a score of 7 reflected high methodological quality (i.e., low risk of bias), a score of 5 or 6 indicated moderate quality, and a score of 4 or less indicated low quality (i.e., high risk of bias).

### 2.7. Statistical Analysis

Mean (±SD) IL-1β levels were extracted to compare between T1DM patients and controls. Since variations in laboratory determination methods were expected (see protocol), the standardized mean difference (SMD) was chosen as an effect size measure, estimated by Cohen’s d method with their corresponding 95% confidence intervals (CI). Data expressed as order statistics (i.e., median, interquartile range and/or maximum–minimum values) were computed and transformed into means (±SD) using the methods proposed by Luo et al. and Wan et al. [20,21]. If it was desirable to combine two or more different means (±SD) from subgroups into a single group, the method provided by the Cochrane Handbook was followed [16]. When the data were only expressed graphically, they were measured and extracted using Engauge Digitizer 4.1. In the meta-analyses, SMDs with 95% CIs were pooled using the inverse-variance method under a random-effects model (based on the DerSimonian and Laird method), which accounts for the possibility that there are different underlying results among study subpopulations (i.e., IL-1β variations among tissues, linked to geographical areas, or related to the inherent heterogeneity of the wide range of experimental methods). Forest plots were constructed to graphically represent the overall effect and for subsequent visual inspection analysis (*p* < 0.05 was considered significant). The heterogeneity between studies was evaluated applying the χ^2^-based Cochran’s Q test (given its low statistical power, *p* < 0.10 was considered significant) and quantified using Higgins I^2^ statistic (values of 50–75% were interpreted as a moderate to high degree of inconsistency across the studies), which estimates what proportion of the variance in observed effects reflects variation in true effects, rather than sampling error [22,23]. Preplanned stratifications (by geographical area, type of tissue, age, Hbg levels, study design, matching, and type of analysis) and univariable meta-regression analyses (by sex and risk of bias) were conducted to identify potential sources of heterogeneity and to explore the potential variation of IL-1β levels on these subgroups [24]. For illustrative purposes, weighted bubble plots were also constructed to graphically represent the fitted meta-regression lines. Sensitivity analyses were additionally performed to test the reliability of our results, evaluating the influence of each individual study on the pooled estimations. For this purpose, the meta-analyses were repeated sequentially, omitting one study each time (the classic “leave-one-out” method). Finally, canonical and contour-enhanced funnel plots were constructed, and the Egger regression test (*p* < 0.10 considered significant) and the non-parametric trim-and-fill method were performed to evaluate small-study effects, such as publication bias [25,26,27,28]. Stata version 16.1 (Stata Corp., College Station, TX, USA) was employed for all tests, with the commands syntax being manually typed (PRG) [29].

## 3. Results

### 3.1. Results of the Literature Search

The flow diagram (Figure 1) depicts the identification and selection process of the studies. We retrieved a total of 3143 records published before October 2020: 626 from MEDLINE/PubMed, 817 from Embase, 826 from the Web of Science, 874 from Scopus, and one [30] from the reference lists screening. After eliminating the duplicates, 1666 studies were considered potentially eligible. After screening their titles and abstracts, 59 were selected for full-text reading. After excluding studies that did not meet all eligibility criteria (all of the studies excluded and their exclusion criteria are listed in the Appendix A), 26 studies were finally included in the systematic review for qualitative evaluation (all of the studies included are listed in the Appendix A) and 25 studies for quantitative meta-analysis. Due to the presence of a considerable degree of clinical, methodological, and statistical heterogeneity, only plasma and serum studies were meta-analyzed to obtain results derived from more homogeneous subpopulations and more reliable results, while determinations from gingival fluid and vitreous humor were omitted from the meta-analysis.

### 3.2. Study Characteristics

Table 1 summarizes the characteristics of the 26 selected studies comparing the changes in circulating IL-1β levels on a total of 4179 T1DM and control patients, and Appendix A exhibits the variables gathered from each study in more detail. One study [31] analyzed IL-1β levels in two tissues (plasma and vitreous humor) being considered as two different analysis units (i.e., n = 27 studies/4233 patients). Sample sizes ranged between 18 and 961 subjects. The studies were conducted in all continents except for Oceania and comprised 12 in Europe, 6 in Asia, 5 in South America, 3 in Africa, and 1 in North America. IL-1β determination was performed by immunoassays in 22 studies (18 by ELISA and 4 by panels; 15 in serum, 5 in plasma, and 1 in gingival crevicular fluid (not meta-analyzed) and 1 in vitreous humor (not meta-analyzed)), flow cytometry in 3 studies (2 in serum and 1 in cord blood plasma), and 2 studies in qRT-PCR (gingival tissue and peripheral blood mononuclear cells (PBMC)).

### 3.3. Qualitative Evaluation

The qualitative analysis was conducted using the Newcastle–Ottawa Scale (NOS), which evaluates potential sources of bias in eight domains (Table 2).

The most frequent biases could be the inadequate description of patient characteristics (age, sex, etc.), failure to report the study period or place of recruitment, and the inclusion of patients outside the population of interest. In our revision, we only included studies in which the groups of diabetic patients were adequately selected and matched between conditions with their respective controls. Studies without a non-T1DM comparator group were excluded.

However, 100% of the studies showed a representativeness of the T1DM patients, selection of the non-T1DM subjects, and proper IL-1β quantification. In relation to confounding factors, the analysis revealed the use of remarkably severe criteria, and no confounding factors were found. Moreover, there were no studies without T1DM patients, improperly diagnosed patients, or insulin-treated patients. The sum of all of these criteria contributes to avoidance of the overall risk of potential bias, increasing the quality of the evidence of the results reported in this systematic review. On the other hand, there were some parameters that introduced a higher possibility of bias. The absence of suitable glycemic control introduces potential bias into our research (30% of the studies were potentially biased). Concerning T1DM progression, this was increased in 46% of the reviewed studies due to the lack of information about the years of evolution of the disease. In regard to the follow up and attrition rate, the risk of bias was elevated in 19% of the studies due to participants being lost to follow-up, which means essential data to evaluate any differences with the characteristics of the final study sample were not fully obtained.

### 3.4. Quantitative Evaluation (Meta-Analysis)

#### 3.4.1. IL-1β Determination by Immunoassays

The *IL-1β* levels were significantly higher in T1DM patients than in controls (SMD = 2.45, 95% CI = 1.73 to 3.17; *p* < 0.001). A significant degree of heterogeneity was observed (*p* < 0.001; I^2^ = 98.6%) (Figure 2, Table 3).

#### 3.4.2. IL-1β Level Determination by Flow Cytometry

The *IL-1β* levels were higher in T1DM patients than in controls (SMD = 1.40, 95% CI = −0.19 to 3.00), close to significant (*p* = 0.08), and very probably underpowered (potentially yielding a non-significant result due to type II error) (n = 3 studies) (Figure 2, Table 3).

#### 3.4.3. IL-1β mRNA Level Determination by qRT-PCR

We did not find significant differences (*p* = 0.59) between T1DM and controls (SMD = −0.66, 95% CI = −3.02 to 1.71). This result was derived from the meta-analysis of only two studies, with imprecise results (very wide confidence intervals) and the true direction of the effect is not yet estimable (Figure 2, Table 3). 

#### 3.4.4. Analysis of Subgroups

Subgroup meta-analyses were only performed for *IL-1β* determination by immunoassays, due to the considerable number of studies (n = 20) and high number of patients (n = 3490) being investigated (Table 3). The statistically significant association was maintained in the following subgroups by continents (Africa: SMD = 10.41, 95% CI = 2.58 to 18.23, *p* = 0.01; Asia: SMD = 2.61, 95% CI = 0.56 to 4.46, *p* = 0.01; Europe: SMD = 1.04, 95% CI = 0.49 to 1.59, *p* < 0.001), age (<18 years: SMD = 2.81, 95% CI = 1.88 to 3.74, *p* < 0.001; >18 years: SMD = 1.56, 95% CI = 0.48 to 2.65, *p* = 0.002), Hbg levels in patients <18 years (Hbg > 7: SMD = 5.43, 95% CI = 3.31 to 7.56, *p* = 0.001), sample source (serum: SMD = 2.73, 95% CI = 1.85 to 3.61, *p* < 0.001; plasma: SMD = 1.34, 95% CI = 0.28 to 2.41, *p* = 0.01), type of analysis (ELISA: SMD = 3.29, 95% CI = 2.27 to 4.30, *p* < 0.001), and study design variables (case-control design: SMD = 2.77, 95% CI = 2.00 to 3.55, *p* < 0.001, age matching: SMD = 3.06, 95% CI= 2.19 to 3.94, *p* < 0.001; sex matching: SMD = 0.55, 95% CI = 0.19 to 0.91, *p* = 0.003) (Table 3) (Appendix A).

#### 3.4.5. Meta-Regression

The potential effect of sex and risk of bias on IL-1β levels determined by immunoassays was explored, but we did not find any significant association for the covariates under analysis (*p* = 0.97 and *p* = 0.91, respectively) (Table 3) (Appendix A).

### 3.5. Quantitative Evaluation (Secondary Analyses)

#### 3.5.1. Sensitivity Analysis

The general results did not substantially vary after the sequential repetition of meta-analyses, omitting one study each time. This suggests that the combined estimations reported do not depend on the influence of a particular individual study (Appendix A).

#### 3.5.2. Small-Study Effects Analysis

These analyses were only applied to the meta-analysis on *IL-1β* determination by immunoassays (n = 22 analysis units). The meta-analyses on *IL-1β* determination by flow cytometry (n = 3) and qRT-PCR (n = 2) harbored low sample sizes, and these methods lack statistical power when the number of primary studies is fewer than ten [28]. Egger’s regression test indicates statistically significant asymmetry (p_Egger_ = 0.014). The funnel plot appears to be slightly asymmetric for the studies plotted at the bottom (Figure 3); however, due to a considerable degree of inter-study heterogeneity, its visual inspection analysis is complex. Consequently, a contour-enhanced funnel plot was constructed (overlaid on the “canonical” funnel plot; Figure 3) to help distinguish publication bias from other causes of asymmetry. This plot leads us to suspect that “missing” studies would be located in the symmetric counterparts with negative significance (i.e., outside of the white region), potentially ruling out publication bias. In addition, the non-parametric trim and fill method did not detect the presence of unpublished studies, confirming the reliability of our results according to the studies published, so the final estimate was not adjusted based on imputation techniques for missing studies.

## 4. Discussion

In this systematic review, significantly higher IL-1β peripheral levels in T1DM patients compared to healthy subjects were shown, according to the meta-analysis on the determination of IL-1β by immunoassays from serum or plasma (SMD = 2.45, 95% CI = 1.73 to 3.17; *p* < 0.001; n = 20 primary-level studies/3490 patients). Young T1DM patients remained significantly higher than T1DM adults. The present study determines an association between glycemic status and IL-1β peripheral levels, which is important in the methodological approach performed to determine them. Based on this window of opportunity, our meta-analysis supports further research on IL-1β as a therapeutic target in T1DM.

The increased peripheral IL-1β levels in childhood indicate a potent role during the first years of the disease, which can contribute to the cytokine storm [32] associated with the first stage of T1DM. Primary prevention strategies targeting inflammatory-mediated comorbidity may prevent secondary complications in the future for these patients [33,34,35]. Previous study revealed the potential therapeutic effects of anti-inflammatory treatments by reducing the peripheral levels of pro-inflammatory cytokines in T1DM-associated complications [36]. Our results suggest that the peripheral pro-inflammatory marker IL-1β is more likely to be increased in the younger population with T1DM compared to adult patients. The different levels detected between the two age ranges studied is particularly significant: child T1DM patients (<18 years old) show a higher IL-1β level than adult T1DM patients (>18 years old), which could be related to the cytokine storming associated with early events in T1DM [37]. Usually, younger T1DM patients present a shorter evolution time of disease, and the immune alterations develop at the beginning of the onset of T1DM. Moreover, in this age group, insulin sensitivity is highly variable due to growth, sexual maturation, and self-care capacity at these ages. In this regard, and supporting these data, young T1DM patients with poor glycemic control present higher IL-1β levels compared with the same age range of T1DM patients with good glycemic management. Several studies have demonstrated an association between the low presence of cytokines and better insulin secretion [12]. The analysis of demographic and geographic area indicates a significant influence of medical assistance and management of T1DM progression care on the level of IL-1β detected.

This study is the first meta-analysis focusing on IL-1β implications in T1DM, different from previous meta-analytic studies based on other cytokines and only restricted to adult patients [12,13,38]. Many previous studies examining other blood cytokine levels during T1DM, such as TNF-α and IL6 [12,13], only included adult patients and this could possibly be attributed by lack of data on IL-1β expression along the T1DM or the inadequate methodological standardization of patient characteristics.

The importance of the methodological approach used for determining IL-1β, and its biological source, has been elucidated in the present work. We found a significantly increased number of analyses performed in serum compared to other biological fluids. The easy accessibility of serum and the periodical clinical testing of it could indicate serum as the principal biological fluid for the determination of inflammatory parameters. Regarding the methodological approaches, we found that the results obtained by ELISA assay are more consistent and with homogeneous groups than other actual immunoassay techniques. The precision of ELISA or the use of only one marker could contribute to a better determination. The exact determination of IL-1β levels is a critical point for determining the clinical standard value, and the results showed in the present analysis confirm that ELISA maintained the range of determination between different studies analyzed; however, other types of immunoassays present a higher range of variability.

IL-1 is a therapeutic target in T1DM patients [11]. Different clinical trials with IL1Ra (anakinra) for adult patients or human monoclonal anti-IL-1β antibody (canakinumab) in pediatric T1DM patients have not been effective in maintaining B-cell function; however, the present meta-analysis showed a critical time point during T1DM progression that could be important to keep in mind for the administration of treatments. A significant relationship between the inflammatory index and β-cell function was not observed in the TN-14 trial. As in consonance with the TN-14 study, our results validated that pediatric onset T1DM is characterized by a more aggressive disease process compared to adult onset T1DM [39], and the relationship identified age dependency in young patients (<18 years (Appendix A)), as shown in the Cabrera et al. article [40]. It is important to note that in the majority of trials, all of the studies focused on evaluating the function of the pancreas using insulin secretion/C-peptide levels; however, in the present study, we tried to elucidate the waves in T1DM-associated IL-1β levels, and determine the relationship with glycated hemoglobin, T1DM management, and age. Usually, the younger T1DM patients present a shorter evolution time of disease, with the immune alterations developing at the beginning of the onset of T1DM. Moreover, in this age group, insulin sensitivity is highly variable due to growth, sexual maturation, and self-care capacity at these ages. In this regard, and supporting these data, the young T1DM patients with a poor glycemic control present higher IL-1β levels compared with the same age range of T1DM patients with good glycemic management. Several studies have demonstrated an association between the low presence of cytokines and better insulin secretion [12]. A positive correlation (change versus change) of plasma HbA1c and plasma IL-6, TNF-α, and IL-1β has been described in a diabetic animal model [41], and in our meta-analysis, we found an association between the glycated hemoglobin and the serum IL-1β levels detected.

According to our qualitative evaluation carried out using the Newcastle–Ottawa Scale, the included studies harbored a low overall risk of potential bias. This fact increases the quality of the evidence of the results reported in our meta-analysis [42]. We also showed that not all studies were conducted, in methodological terms, with the same rigor. Studies should more meticulously communicate the years of evolution of the disease, and control groups should be more carefully designed, being appropriately matched for age and sex. Future studies assessing the relationships between IL-1β levels among T1DM patients could consider the recommendations given in this systematic review and meta-analysis to improve and standardize future research.

Some potential limitations should also be discussed. First, our meta-analysis revealed a considerable degree of inter-study heterogeneity. Heterogeneity is a common finding in meta-analyses dealing with serum biomarkers—particularly cytokines—measured and expressed as continuous variable [12,13]. It must also be noted that a random-effects model was applied in all meta-analyses to account for heterogeneity. When considering the uses and limitations of meta-analytical techniques, a key strength is the ability to reveal patterns across the study results and identifying potential subpopulations (i.e., sources of heterogeneity) [43]. In this sense, our meta-analysis may have identified differences among geographical regions, age, Hbg levels, and analysis techniques, among other factors, that may constitute true sources of heterogeneity, potentially exerting an impact on IL-1β level variations in T1DM. Furthermore, only plasma and serum determinations were meta-analyzed to obtain results derived from more homogeneous clinical and methodological subgroups. Future studies are needed to obtain a higher quality of evidence on the determinations derived from other anatomical sites (e.g., crevicular gingival fluid or vitreous humor). Another element that can explain the heterogeneity is the lack of standardization of the assays used to measure IL-1β. Second, visual inspection analysis of the canonical funnel plot and statistical analyses detected the presence of asymmetry, pointing out small-study effects. Therefore, the random-effects model could be overestimating our results, giving more weight to the studies with a lower sample size, where sampling error may be influential [44]. Nevertheless, the enhanced-contour funnel plot and the trim and fill method allowed us to suspect that the reported asymmetry is artefactual, due to sampling variation or to chance [9], and not really to the presence of publication bias, which could be ruled out [26]. Third, another potential limitation could be related to our eligibility criteria, where clinical trials were excluded, in spite of the advantages in longitudinal associations derived from this study design. In order to meet our objectives, we first a priori designed our study protocol, and we only considered primary-level cohort studies/small case series, case control, and cross-sectional studies to be included, due to their observational study design. There are controversies on the integration of observational and interventional mixed primary-level studies in meta-analysis, particularly in the context of molecular biomarkers with clinical implications. Since our research was performed to better understand the natural history of the condition type 1 diabetes mellitus in the context of IL-1β levels, the inclusion of treatment/interventionist studies (which, by definition, try to decrease the chronic inflammation in diabetes or to eliminate risk factors) could potentially distort the reality of this disease, attenuate inflammation, modulate il-1β levels, introduce a new heterogeneity source, and, consequently, affecting the achievement of our goals. Finally, another potential limitation is the absence of an association between secondary complications, such as diabetic retinopathy, and the IL-1β levels. It was demonstrated that IL-1β increased in the diabetic mouse retina and IL-1β induced pericyte apoptosis via NF-κB activation under high glucose conditions, thereby increasing endothelial permeability in diabetic retinopathy [45]. However, we could not undertake a meta-analytical approach on the diabetic secondary complications due to the low number of articles with inclusion criteria established. Despite the above limitations, the study strengths include our careful study design, a sensitive literature search strategy, the absence of restrictions by date limits or publication language, robust qualitative recommendations for the development and design of future studies on this topic, and the comprehensive meta-analytical approach, showing powerful statistical findings across many analyses.

## 5. Conclusions

In conclusion, this systematic review and comprehensive meta-analysis provides a deep exploration of the possible role of IL-1β as a tool cytokine in T1DM progression and management of disease. IL-1β is significantly increased in young T1DM patients, which can be used as a marker to initiate the administration of new therapeutic approaches for IL-1β modulation. The relationship between the status of T1DM and IL-1β levels measured by ELISA corroborate the strong affinity between the inflammatory context and T1DM glycemic status, determined by Hbg levels. Further analysis and validation are needed to establish a clinical standard value for IL-1β associated with different T1DM status. The results obtained allow for the hypothesis of a potential role of IL-1β as a therapeutic target in the early stages of T1DM, where the actual treatments are focused on the pharmacological abrogation of IL-1β action and reducing T1DM progression. The evaluation of IL-1β levels in the early stages of the disease could support the finding that inflammatory status is associated with glycemic control.

## Figures and Tables

**Figure 1 jcm-11-01303-f001:**
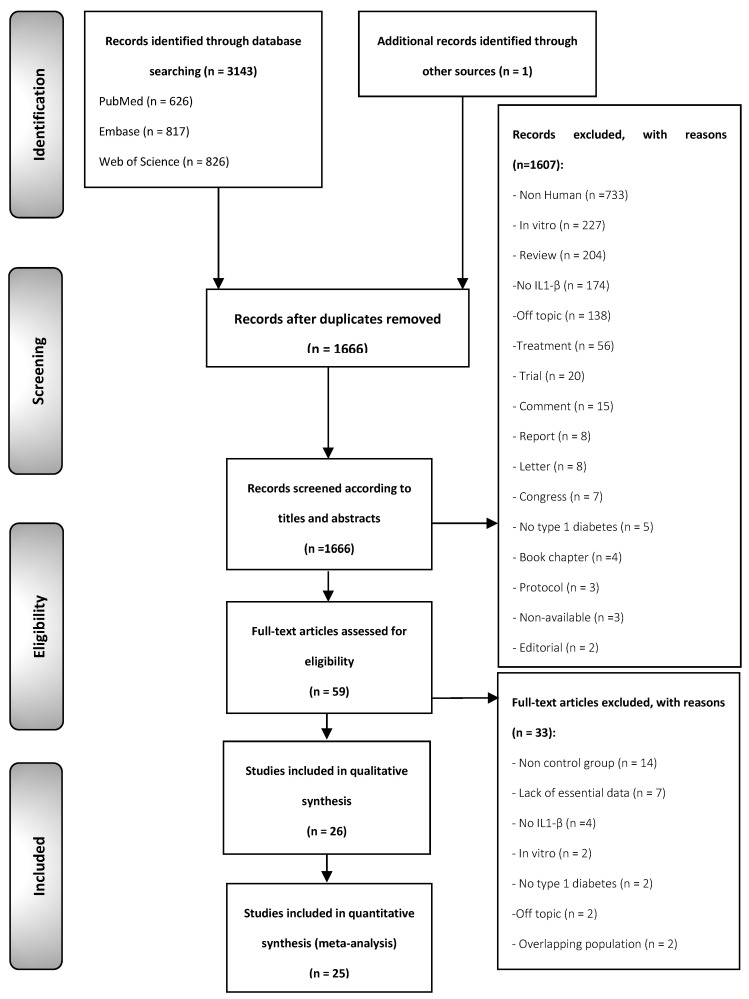
Flow diagram. Identification and selection process of relevant studies comparing IL-1β levels between T1DM patients and controls.

**Figure 2 jcm-11-01303-f002:**
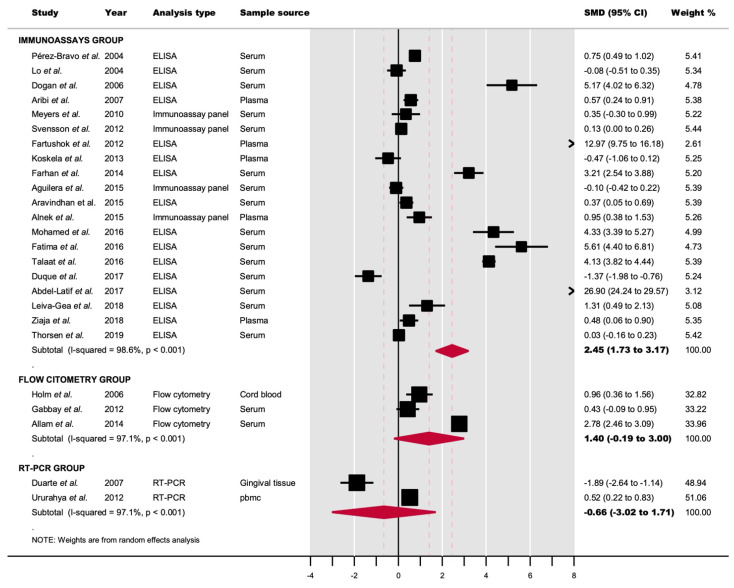
Forest plot graphically representing the meta-analyses evaluating the changes in circulating IL-1β levels between T1DM patients and controls (random-effects models, inverse-variance weighting based on the DerSimonian and Laird method). Standardized mean difference (SMD) was chosen as effect size measure. An SMD > 0 suggests that IL-1β levels are higher in T1DM. Diamonds indicate the overall pooled SMDs with their corresponding 95% confidence intervals (CI).

**Figure 3 jcm-11-01303-f003:**
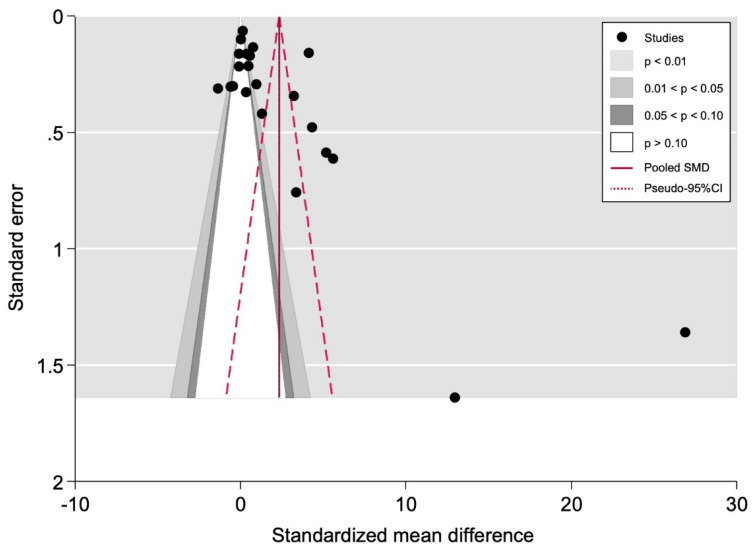
Canonical and contour funnel plots of the estimated circulating IL-1β levels (assessed across immunoassays) comparing type 1 diabetes mellitus and controls, expressed as standardized mean difference (SMD) against its standard error. The red vertical line corresponds to the pooled SMD estimated in the meta-analysis. The two diagonal intermittent lines represent their pseudo-95% CI. Contours represent the defined conventional levels of statistical significance (i.e., 0.01, 0.05, 0.10) accompanied by associated shaded regions. The black circles represent the 22 studies meta-analyzed.

**Table 1 jcm-11-01303-t001:** Summarized characteristics of reviewed studies.

Total	26 studies *
Year of publication	2004–2019
Number of patients	
Total	4179 patients *
Cases with T1DM	2186 patients
Controls	2047 patients
Sample size, range	18–961 patients
IL-1β determination	
Immunoassays	22 studies (18 by ELISA, 4 by panels)
Flow cytometry	3 studies
qRT-PCR	2 studies
Source of samples	
Serum	17 studies
Plasma	5 studies
Gingival crevicular fluid	1 study
Vitreus humour	1 study
Cord blood plasma	1 study
Gingival tissue	1 study
Peripheral blood leukocytes	1 study
Geographical region	
Europe	12 studies
Asia	6 studies
South America	5 studies
Africa	3 study
North America	1 study

*—One study (Koskela et al., 2013) analyzed IL-1β levels in two tissues (plasma and humour vitreus), being considered as two different analysis units (i.e., n total = 27 studies/4233 patients).

**Table 2 jcm-11-01303-t002:** Summary of risk of bias assessment based on Newcastle–Ottawa Quality Assessment Scale. Two reviewers who had content and methodological expertise independently and in duplicate assessed and graded the risk of bias for the included studies with an adapted version of the Newcastle–Ottawa scale (NOS), which has been described elsewhere [8]. The assessments were compared and conflicts resolved by agreement between the two reviewers. The maximum score was 8, the minimum score 0. It was decided a priori that a score of 7 was reflective of high methodological quality (e.g., low risk of bias), a score of 5 or 6 indicated moderate quality, and a score of 4 or less indicated low quality (e.g., high risk of bias). A filled blue star indicates that a star has been awarded, and a blank star indicates that no star has been awarded and the study has been graded as poor quality in that category [8]. Wells GA (2010) The Newcastle–Ottawa Scale (NOS) For Assessing The Quality Of Non Randomised Studies In Meta-Analyses. Ottawa (ON): Ottawa Health Research Institute.

Study	Selection	Control	Outcomes	Overall Quality
	Representativ eness of the T1DM patients	Selection of the non-T1DM subjects	Properly IL1b quantification	Glycemic control	Control of confounding factors	Assessment of T1DM progression	Appropriate follow up period	Adequacy of follow up	
Pérez-Bravo et al. (2004)						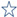			High
Lo et al. (2004)						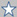			High
Holm et al. (2006)									High
Dogan et al. (2006)				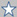					High
Arabi et al. (2007)									High
Duarte et al. (2007)									High
Salvi et al. (2010)						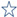			High
Meyers et al. (2010)									High
Gabbay et al. (2012)				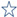		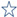			Moderate
Svensson et al. (2012)				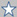					High
Ururahy et al. (2012)									High
Fartushok et al. (2012)						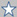			High
Koskela et al. (2013)						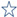			High
Allam et al. (2014)									High
Farhan et al. (2014)				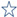		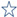			Moderate
Aguilera et al. (2015)								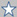	High
Aravindhan et al. (2015)				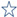				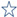	Moderate
Alnek et al. (2015)									High
Mohamed et al. (2016)				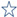					High
Fatima et al. (2016)						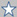		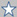	Moderate
Talaat et al. (2016)									High
Duque et al. (2017)				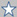		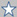		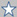	Moderate
Abdel-Latif et al. (2017)						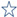			High
Leiva-Gea et al. (2018)									High
Ziaja et al. (2018)				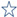		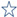		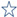	Moderate
Thorsen et al. (2019)						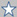			High

**Table 3 jcm-11-01303-t003:** Meta-analyses on circulating IL-1β levels in type 1 diabetes mellitus.

					Pooled Data		Heterogeneity		
Meta-Analyses	No. of Studies	No. of Patients	Stat. Model	Wt	SMD (95% CI)	*p-Value*	*P_het_*	*I*^2^ (%)	* Appendix A ^a^ *
**Determination by immunoassays**									
All ^b^	20	3490	REM	D-L	2.45 (1.73 to 3.17)	<0.001	<0.001	98.6	——
Subgroup analysis by geographical area ^c^									Appendix A
Africa	3	403	REM	D-L	10.41 (2.58 to 18.23)	0.01	<0.001	99.5	
Asia	5	885	REM	D-L	2.61 (0.56 to 4.66)	0.01	<0.001	99.0	
Europe	9	1875	REM	D-L	1.04 (0.49 to 1.59)	<0.001	<0.001	95.0	
North America	1	38	——	——	0.35 (−0.30 to 0.99)	0.29	——	——	
South America	2	289	REM	D-L	−0.29 (−2.37 to 1.79)	0.78	< 0.001	97.4	
Subgroup analysis by age ^c^									Appendix A
<18 years old	14	2870	REM	D-L	2.81 (1.88 to 3.74)	<0.001	<0.001	98.9	
>18 years old	6	620	REM	D-L	1.56 (0.48 to 2.65)	0.002	<0.001	96.5	
Subgroup analysis by HbAc1 levels in patients <18 years old ^c,d^									Appendix A
<7	2	79	REM	D-L	−0.04 (−2.67 to 2.58)	0.97	<0.001	96.2	
>7	8	1138	REM	D-L	5.43 (3.31 to 7.56)	0.001	<0.001	99.1	
Subgroup analysis by age matching ^c^									Appendix A
Matched	15	3172	REM	D-L	3.06 (2.19 to 3.94)	<0.001	<0.001	98.8	
Unmatched	5	318	REM	D-L	0.90 (−0.18 to 1.97)	0.10	<0.001	94.4	
Subgroup analysis by sex matching ^c^									Appendix A
Matched	11	2379	REM	D-L	0.55 (0.19 to 0.91)	0.003	<0.001	92.9	
Unmatched	3	224	REM	D-L	0.88 (−1.15 to 2.90)	0.40	<0.001	97.5	
NA	6	887	REM	D-L	8.66 (5.37 to 11.96)	<0.001	<0.001	98.9	
Subgroup analysis by sample source ^c^									Appendix A
Serum	15	3111	REM	D-L	2.73 (1.85 to 3.61)	<0.001	<0.001	98.9	
Plasma	5	379	REM	D-L	1.34 (0.28 to 2.41)	0.01	<0.001	94.3	
Subgroup analysis by type of analysis ^c^									Appendix A
ELISA	16	2235	REM	D-L	3.29 (2.27 to 4.30)	<0.001	<0.001	98.8	
Immunoassay panel	4	1255	REM	D-L	0.25 (−0.08 to 0.58)	0.14	0.02	70.5	
Subgroup analysis by study design ^c^									Appendix A
Case-control	16	2447	REM	D-L	2.77 (2.00 to 3.55)	<0.001	<0.001	98.1	
Cohort	1	398	—	—	0.03 (−0.164 to 0.23)	0.74	—	—	
Cross-sectional	3	645	REM	D-L	1.39 (−1.56 to 4.34)	0.36	<0.001	99.3	
Univariable meta-regression ^e^									
Sex (% of T1DM males)	17	2928	Random-effects Meta-regression		Coef = 0.011 (−0.619 to 0.641)	0.97	——	——	Appendix A
Risk of bias (NOS score)	20	3490	Random-effects Meta-regression		Coef = 0.195 (−3.209 to 3.598)	0.91	——	——	Appendix A
**Determination by qRT-PCR**									
All ^b^	2	216	REM	D-L	−0.66 (−3.02 to 1.71)	0.59	<0.001	97.1	——
**Determination by Flow Citometry**									
All ^b^	3	455	REM	D-L	1.40 (−0.19 to 3.00)	0.08	<0.001	91.8	——

Abbreviations: Stat., statistical; Wt, method of weighting; SMD, standardized mean difference; CI, confidence intervals; REM, random-effects model; D-L, DerSimonian and Laird method; HbAc1, hemoglobin Ac1; T1DM, type 1 diabetes mellitus; NOS, Newcastle–Ottawa Scale; NA, not available. ^a^ More information in the Appendix A; ^b^ meta-analyses; ^c^ subgroup meta-analyses; ^d^ the studies recruiting patients >18 years old or with missing data were excluded for this analysis; ^e^ effect of study covariates on circulating IL-1β levels among patients with T1DM compared with controls.

## Data Availability

The data that support the findings of this study are available in the Appendix A of this article.

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
