# Peer review of "IL-1β Implications in Type 1 Diabetes Mellitus Progression: Systematic Review and Meta-Analysis"

_jcm, 2022, doi:10.3390/jcm11051303_

Round 1

Reviewer 1 Report

In principal, interesting topic of the manuscript, but 

1.) The paper is very hard to read and to understand because there are a lot of grammatical and spelling errors. Several word are missing at all.

2.) In the analysis of the different included paper there is only a separation according to determination of IL-1ß by immunoassays, flow citometry and RT-PCR. Only in the immunoassay studies a significant higher concentration of IL-1ß could be shown in persons with Diabetes mellitus Typ 1 (Figure 2.) A problem seems from my point of view that also 2 determinations of IL-1ß in the ginigival fluid and in the virteous homour were included whereas all others came from serum or plasma. The 2 single studies should be omitted from the meta-analysis because otherwise it is too heterogenous and no conclusion can be drawn. Is there a methodical difference between concentarions of IL-1ß in serum or plasma ?

3.) Moreover, I do not quite understand what the authors were looking for. The possible role of IL-1ß in DM Typ 1 in

early stages of the disease and manifestations (first diagnosis) as a potential pathophysiological target for slowing down ß-cell destruction by drugs (f.e specific IL-1ß antibodies or blockers which has been published in small case-series) ?

or in late stage of disease as a marker or target for development of diabetic late complication (different forms of diabetic micro- or macrangiopathy ?)

Is there a possible correlation of IL-1ß with duration of disease, age or quality of metabolic control (f.e. measured by acute glycemia or HbA1c ?)

Unless this questions are not clarified, no concrete message is possible from the shown results of the meta-analysis,

Author Response

Reviewer 1

Comment 1

In principal, interesting topic of the manuscript

Response 1

We are very grateful to Reviewer 1 for his/her kind comments. All the reviewer's instructions have been considered and are responded to below.

Comment 2

The paper is very hard to read and to understand because there are a lot of grammatical and spelling errors. Several words are missing at all.

Response 2

Following the comments of both reviewers, the manuscript has been sent to a professional translator (English editing at https://www.mdpi.com/authors/english) to be revised and corrected. Language and formatting, especially grammatical and spelling errors, have been carefully revised and improved throughout the entire manuscript. All changes were included using the “Track Changes” function in the newly revised version of the manuscript.

Comment 3

In the analysis of the different included papers there is only a separation according to determination of IL-1ß by immunoassays, flow cytometry and RT-PCR. Only in the immunoassay studies a significant higher concentration of IL-1ß could be shown in persons with Diabetes mellitus Type 1 (Figure 2.) A problem seems from my point of view that also 2 determinations of IL-1ß in the gingival fluid and in the vitreous humour were included whereas all others came from serum or plasma. The 2 single studies should be omitted from the meta-analysis because otherwise it is too heterogenous and no conclusion can be drawn.

Response 3

Thank you for this important comment. Following the reviewer’s comments, the meta-analysis on immunoassays studies, associated subgroup meta-analyses (i.e., stratified by geographical area, age, sex and age matching, sample source, type of analysis, and study design), and meta-regressions (by sex and risk of bias) was updated, omitting the determinations from gingival fluid and vitreous humors to reduce clinical and methodological heterogeneity. Consequently, the following changes and new paragraphs have been updated or newly added:

Abstract:

“Compared with controls, IL-1β determined by immunoassays (pooled standardized mean difference (SMD)=2.45, 95% CI=1.73 to 3.17; p<0.001) was significantly elevated in T1DM. Compared with controls, IL-1β levels in patients <18 years (SMD=2.81, 95% CI=1.88-3.74, p<0.001) with higher hemoglobin-glycated (Hbg) levels (SMD=5.43, 95% CI=3.31-7.56; p=0.001) were respectively significantly elevated. Compared with the study design, IL-1β evaluated by ELISA (pooled SMD=3.29, 95% CI=2.27 to 4.30, p<0.001) was significantly elevated in T1DM patients.”

Results of the literature search section:

“…Due to the presence of a considerable clinical, methodological, and statistical heterogeneity degree, only plasma and serum studies were meta-analyzed to obtain results derived from more homogeneous subpopulations and more reliable results, while determinations from gingival fluid and vitreous humors were omitted from the meta-analysis.”

IL-1ß determination by immunoassays results section:

“IL-1ß levels were significantly higher in T1DM patients than in controls (SMD=2.45, 95% CI=1.73 to 3.17; p<0.001). A considerable significant degree of heterogeneity was observed (p<0.001; I2=98.6%) (Fig. 2, Table 3).”

Analysis of subgroups results section:

“Subgroup meta-analyses were only performed for IL-1ß determination by immunoassays due to a great number of studies (n=20) and a high number of patients (n=3,490) investigated (Table 3). The statistically significant association was maintained in the following subgroups by continents (Africa: SMD=10.41, 95% CI=2.58 to 18.23, p=0.01; Asia: SMD=2.61, 95% CI=0.56 to 4.46, p=0.01; Europe: SMD=1.04, 95% CI=0.49 to 1.59, p<0.001), age (<18 years: SMD=2.81, 95% CI=1.88 to 3.74, p<0.001; >18 years: SMD=1.56, 95% CI=0.48 to 2.65, p=0.002), Hbg levels in patients <18 years (Hbg>7: SMD=5.43, 95% CI=3.31 to 7.56, p=0.001), sample source (serum: SMD=2.73, 95% CI=1.85 to 3.61, p<0.001; plasma: SMD=1.34, 95% CI=0.28 to 2.41, p=0.01, type of analysis (ELISA: SMD=3.29, 95% CI=2.27 to 4.30, p<0.001), and study design variables (case–control design: SMD=2.77, 95% CI=2.00 to 3.55, p<0.001, age matching: SMD=3.06, 95% CI= 2.19 to 3.94, p<0.001; sex matching: SMD=0.55, 95% CI=0.19 to 0.91, p=0.003) (Table 3) (Fig. S1-8, Appendix pp.11-18).”

Meta-regression results section:

“The potential effect of sex and risk of bias on IL-1β levels determined by immunoassays was explored, but we did not find any significant association for the covariates under analysis (p=0.97 and p=0.91, respectively) (Table 3) (Fig. S9, 10, Appendix pp.19,20)”.

Discussion’s first paragraph:

“In this systematic review, significantly higher IL-1β peripheral levels in T1DM patients compared to healthy subjects were shown, according to the meta-analysis on the determination of IL-1ß by immunoassays from serum or plasma (SMD=2.45, 95% CI=1.73 to 3.17; p<0.001; n=20 primary-level studies/3,490 patients).”

Limitations:

“…Furthermore, only plasma and serum determinations were meta-analyzed to obtain results derived from more homogeneous clinical and methodological subgroups. Future studies are needed to obtain a higher quality of evidence on determinations derived from other anatomical sites (e.g., crevicular gingival fluid or vitreous humors).”

Finally, the associated tables and figures were also updated in the manuscript (i.e., Figure 1/flow diagram, Figure 2/forest plot, and Table 3/meta-analytical results) and in the Appendix (supplementary forest plots from subgroups meta-analyses (Fig. S1, S2, S4-S8), bubble plots from meta-regression analyses (Fig. S9 and S10), and the sensitivity analysis (Table S3)).

Comment 4

Is there a methodical difference between concentrations of IL-1ß in serum or plasma?

Response 4:

Thank you very much for your comment. Usually, the detection assays for IL-1b indicate if its product is optimized for plasma or serum. However, the detection assays are availed for both types of samples, and the standard curve used is the same for both types of samples. The most important difference between both biological fluids' detection is the methodological approach used.

The present study has considered the fold change between the groups, so the consistent results obtained have been reinforced by the similarity IL-1b profile in plasma or serum samples.

Comment 5a

Moreover, I do not quite understand what the authors were looking for. The possible role of IL-1ß in DM Type 1 in early stages of the disease and manifestations (first diagnosis) as a potential pathophysiological target for slowing down ß-cell destruction by drugs (f.e specific IL-1ß antibodies or blockers which has been published in small case-series)?

Response 5a

Thanks in advance for your recommendation. Recently, many studies have given attention to the role of IL-1b in the pathogenesis of autoimmune and inflammatory diseases. The role of IL-1β during T1DM has been the critical point of study in the last years. There are numerous studies that relate the polymorphisms and gene variations in the IL-1β gene with the differences in the transcription and expression of the IL-1β gene that could be correlated with many autoimmune and inflammatory diseases' development. In this point, the early detection of IL-1β genetic alterations or increased serum/plasma levels could be a critical step to determine T1DM evolution and management of the disease.

On the other hand, studies have demonstrated that the genetic or pharmacological abrogation of IL-1β action reduces disease incidence in animal models of T1DM. In this regard, the actual clinical trials are focused on modulating the effect of IL-1β, being a potential interventional target in autoimmune diabetes mellitus.

Comment 5b

or in late stage of disease as a marker or target for development of diabetic late complication (different forms of diabetic micro- or macro-angiopathy ?)

Response 5b

It was demonstrated that increased IL-1β in the diabetic mouse retina and IL-1β induced pericyte apoptosis via NF-κB activation under high glucose conditions, thereby increasing endothelial permeability in diabetic retinopathy (Biochem Biophys Res Commun. 2021 Mar 26;546:46-53. doi: 10.1016/j.bbrc.2021.01.108. Epub 2021 Feb 8.). However, in our study, we could not include the articles related to secondary diabetic complications due to the low number of primary-level studies with inclusion criteria established. In this regard, we have included this comment as a possible limitation in our study.

Comment 5c

Is there a possible correlation of IL-1ß with duration of disease, age or quality of metabolic control (f.e. measured by acute glycemia or HbA1c ?)

Response 5c

Thank you very much for your comment. As the review suggests, there is a potential association between IL-1β levels during T1DM at early stages of the disease, when the metabolic status is poor, as we have shown in the analysis of subgroups (new Table 3 and new Fig. S1-8, Appendix pp.11-18). T1DM is an autoimmune disease that appears during childhood, but the results have demonstrated that IL-1β levels are higher in young patients with a shorter T1DM duration than older patients with more years of T1DM duration. This explication could be supported by the quality of metabolic control (represented in HbA1c levels).

Comment 6

Unless these questions are not clarified, no concrete message is possible from the shown results of the meta-analysis

Response 6:

Dear reviewer, thank you again for your effort, helpful comments, and the opportunity to improve the quality and scientific level of this research. The manuscript has been thoroughly rewritten and completely modified, and in our opinion, all questions were clarified.

Our systematic review and meta-analysis establish a relation in IL-1β levels during T1DM at early stages that could be responsible for a proinflammatory environment being detected. This deleterious status co-exists in the presence of poor metabolic control, which is usually present during the onset of T1DM in childhood.

Reviewer 2 Report

I have read with interest the review of Dr. Cano-Cano and colleagues. The work is comprehensive and offers a clear insight as to how to move forward with respect to IL1B's inflammatory role in T1DM. I only have a few comments.

  1. The introduction can be improved as not much background information were mentioned about IL1B. The authors should also explain the rationale behind the review as to why IL1B is an emerging biomarker for inflammation in autoimmune diseases such as T1DM.
  2. Another element that can explain the heterogeneity is the lack of standardization of assays used to measure IL1B. The authors may wish to expound on this.
  3. Why were clinical trials excluded? This could have offered longitudinal associations between T1DM progression and altered IL1B.
  4. The conclusion is rather weak and the authors may add recommendations as to how to move forward and what's next for IL1B.
  5. The authors may need the assistance of a native English-speaker to proofread the work prior to resubmission. 

Author Response

Reviewer 2

Comment 1

I have read with interest the review of Dr. Cano-Cano and colleagues. The work is comprehensive and offers a clear insight as to how to move forward with respect to IL1B's inflammatory role in T1DM. I only have a few comments.

Response 1

We are also very grateful to Reviewer 2 for his/her kind comments.

Comment 2

The introduction can be improved as not much background information were mentioned about IL1B. The authors should also explain the rationale behind the review as to why IL1B is an emerging biomarker for inflammation in autoimmune diseases such as T1DM.

Response 2:

Thank you very much for your suggestion. As the reviewer has recommended, the Introduction has been modified, including more background information about IL-1β role as a biomarker or therapeutic target in new experimental approaches. Please note that the text has been changed using the “Track Changes” function.

Comment 3

Another element that can explain the heterogeneity is the lack of standardization of assays used to measure IL1B. The authors may wish to expound on this.

Response 3

Following the reviewer’s advice, this sentence has been now included in the Limitations section:

“Another element that can explain the heterogeneity is the lack of standardization of assays used to measure IL-1β”.

Comment 4

Why were clinical trials excluded? This could have offered longitudinal associations between T1DM progression and altered IL1B.

Response 4

Dear reviewer, thank you for this important comment. To meet our objectives, we first a priori designed our study protocol, where, as the reviewer rightly has commented, we only considered primary-level cohort studies/small case series and case–control studies to be included due to their observational study design. There are controversies on integrating observational and interventional mixed studies in meta-analyses, specifically in molecular biomarkers with clinical implications. Since our objective was to better understand the natural history of type 1 diabetes mellitus in the context of IL-1β levels, the inclusion of treatment/interventionist studies (which by definition should try to decrease diabetes chronic inflammation or eliminate risk factors) could potentially distort the reality of this disease, attenuate inflammation, modulate IL-1β levels, introduce a new heterogeneity source (the famous quote in meta-analysis “the risk is to mix apples with oranges”), and consequently, our goals could not be achieved. Nevertheless, this important comment has also been included in the Limitations paragraph, justifying our eligibility criteria:

“… Another potential limitation could be related to our eligibility criteria, where clinical trials were excluded, despite the advantages in longitudinal associations derived from this study design. To meet our objectives, we first a priori designed our study protocol, and we only considered primary-level cohort studies/small case series, case–control, and cross-sectional studies to be included due to their observational study design. There are controversies on integrating observational and interventional mixed primary-level studies in meta-analysis, specifically in molecular biomarkers with clinical implications. Since our research was performed to better understand the natural history of type 1 diabetes mellitus in the context of IL-1β levels, the inclusion of treatment/interventionist studies (which by definition try to decrease the early chronic inflammation in T1DM or to eliminate risk factors) could potentially distort the reality of this disease, attenuate inflammation, modulate IL-1β levels, introduce a new heterogeneity source, and consequently, our goals could not be achieved.”

Furthermore, one of the study inclusion criteria was defined by a control or healthy group, and it is important to note that clinical trials have usually been performed with placebo or treatment conditions.

Comment 5

The conclusion is rather weak and the authors may add recommendations as to how to move forward and what's next for IL1B.

Response 5

Thank you very much for your comment. We have improved the Conclusions section following your suggestions. The text below has been included.

“The results obtained allow us to hypothesize a potential role of IL-1β as a therapeutic target in early stages of T1DM, where the actual treatments are focused in pharmacological abrogation of IL-1β action and reducing T1DM progression. The evaluation of IL-1β levels in early steps of disease could support that inflammatory status is next to glycemic control”.

Comment 6

The authors may need the assistance of a native English-speaker to proofread the work prior to resubmission.

Response 6

Following the comments of both reviewers, the manuscript has been sent to a professional translator (English editing at https://www.mdpi.com/authors/english.) to be revised and corrected. Language and formatting, specifically grammatical and spelling errors, have been carefully revised and improved throughout the entire manuscript. All changes were included using the "Tracked Changes" function in the newly revised version of the manuscript.

Round 2

Reviewer 1 Report

Sufficient revision

Reviewer 2 Report

I commend the authors for the substantial revisions done. They have fully addressed my comments.